# Non-Invasive Spectroscopy for Measuring Cerebral Tissue Oxygenation and Metabolism as a Function of Cerebral Perfusion Pressure

**DOI:** 10.3390/metabo12070667

**Published:** 2022-07-20

**Authors:** Deepshikha Acharya, Ankita Mukherjea, Jiaming Cao, Alexander Ruesch, Samantha Schmitt, Jason Yang, Matthew A. Smith, Jana M. Kainerstorfer

**Affiliations:** 1Department of Biomedical Engineering, Carnegie Mellon University, Pittsburgh, PA 15213, USA; dacharya@andrew.cmu.edu (D.A.); amukherjea310@gmail.com (A.M.); jiamingc@andrew.cmu.edu (J.C.); sammi@cmu.edu (S.S.); jasony1@andrew.cmu.edu (J.Y.); mattsmith@cmu.edu (M.A.S.); 2Neuroscience Institute, Carnegie Mellon University, Pittsburgh, PA 15213, USA; aruesch@andrew.cmu.edu

**Keywords:** cerebral metabolic rate of oxygen, cerebral tissue oxygenation, cerebral perfusion pressure, cerebral autoregulation, diffuse optics

## Abstract

Near-infrared spectroscopy (NIRS) and diffuse correlation spectroscopy (DCS) measure cerebral hemodynamics, which in turn can be used to assess the cerebral metabolic rate of oxygen (CMRO_2_) and cerebral autoregulation (CA). However, current mathematical models for CMRO_2_ estimation make assumptions that break down for cerebral perfusion pressure (CPP)-induced changes in CA. Here, we performed preclinical experiments with controlled changes in CPP while simultaneously measuring NIRS and DCS at rest. We observed changes in arterial oxygen saturation (~10%) and arterial blood volume (~50%) with CPP, two variables often assumed to be constant in CMRO_2_ estimations. Hence, we propose a general mathematical model that accounts for these variations when estimating CMRO_2_ and validate its use for CA monitoring on our experimental data. We observed significant changes in the various oxygenation parameters, including the coupling ratio (CMRO_2_/blood flow) between regions of autoregulation and dysregulation. Our work provides an appropriate model and preliminary experimental evidence for the use of NIRS- and DCS-based tissue oxygenation and metabolism metrics for non-invasive diagnosis of CA health in CPP-altering neuropathologies.

## 1. Introduction

Healthy brain function requires a tight coupling between the cerebral oxygen demand and cerebral oxygen supply. Oxygen demand is usually linked to underlying neural activity, reflected in the cerebral metabolic rate of oxygen (CMRO_2_). The oxygen supply depends on the cerebral blood flow (CBF), which is modulated by the cerebral perfusion pressure (CPP). While CPP alters with variations in the mean arterial pressure (MAP) or intracranial pressure (ICP), a healthy brain can ensure stable CBF over a wide range of CPP through a mechanism called cerebral autoregulation [1,2]. In several cerebral pathologies, CPP can alter drastically, impairing CBF regulation. The consequent disruption in the oxygen supply can adversely affect CMRO_2_. In such cases, ensuing secondary brain damage can worsen patient outcomes [3,4,5]. Thus, monitoring of CMRO_2_ and CBF coupling, often referred to as the coupling ratio, is an important aspect of clinical treatment plans [6,7,8].

Current clinical standards for measuring CMRO_2_ and CBF require invasive monitors to be placed directly in the brain. However, such monitors are surgically placed only for acute cases and can only monitor parameters at a single cerebral location. This can restrict the capacity for timely diagnosis in patients, potentially worsening prognosis. Recent developments in real-time non-invasive monitoring techniques provide a great alternative to current clinical standards. Specifically, diffuse optical techniques such as near-infrared spectroscopy (NIRS) and diffuse correlation spectroscopy (DCS) have the benefit of providing a bedside-compatible non-invasive alternative with the ability to directly measure cerebral tissue oxygenation [9].

Frequency-domain NIRS can measure absolute [10] and relative changes in the oxygenated and deoxygenated hemoglobin concentration [11] in tissue. Consequently, this can help estimate tissue oxygen saturation of hemoglobin (StO_2_) and arterial oxygen saturation (SaO_2_). Additionally, DCS can be used to measure relative changes in CBF [12]. These parameters can then be used to calculate CMRO_2_. Multiple mathematical models [13,14,15,16,17] have been described over the years, and the nuance of each model resides in the assumptions and simplifications used, affecting its practical applicability. One commonly used model [13] assumes a constant SaO_2_ and arterio-venous volume fraction (referred to as gamma, γ), which works well under the specific condition of negligible CPP variation, as expected in most studies with healthy participants. This model, however, may fail when monitoring subjects with autoregulatory failure and extreme CPP.

Here, we discuss a general model proposed by Kocsis et al. [16] for CMRO_2_ estimation, which specifically accounts for variations in SaO_2_ and arterial blood volume changes with CPP. We tested its applicability in estimating CMRO_2_, among other tissue oxygenation metrics, during controlled changes in CPP and, therefore, cerebral autoregulation (CA), with data collected from four non-human primates. We contrasted the results from this general model with the alternative model for CMRO_2_ estimation with the constant γ and constant SaO_2_ assumption, highlighting its shortcomings specifically when studying a wide range of CPP. Finally, we validated the use of NIRS- and DCS-based measures with our general model to provide a multi-parameter diagnosis of cerebral autoregulatory health, adding to the strong evidence in their favor for bedside non-invasive monitoring of neurocritical patients.

## 2. Results

We recorded continuous hemodynamic signals (on average 10 h per subject) using NIRS and DCS in four anesthetized non-human primates in a series of controlled experiments to study changes in cerebral tissue oxygenation and metabolism with CPP, linked to CA and its failure. CPP variations were induced by gradually changing the fluid pressure in the ventricle. Blood gas changes affecting CA [18] were controlled for by externally ventilating the subjects and maintaining end-tidal CO_2_ (EtCO_2_). The experimental setup and time traces of ICP, CPP, and EtCO_2_ from an example subject are shown in Figure 1a and b, respectively.

Here, we discuss trends and results in different tissue oxygenation and metabolic metrics recorded with DCS and NIRS over a CPP range of 40–160 mmHg. Further, we compare the results using two different methods of CMRO_2_ estimation: the general method by Kocsis et al. [16] and the constant gamma model by Culver et al. [13]. Finally, we use the former model to provide some biomarkers for bedside non-invasive monitoring of CA health.

### 2.1. Changes in Tissue Oxygenation with Autoregulation

Various tissue oxygenation and metabolic metrics are shown in Figure 2 as the group averages across CPP of 2 mmHg bins, across all the subjects. All quantities are indicated with a prefix ‘r’ to show the value of a parameter relative to its baseline value. The baseline for each parameter was calculated as an average over the first 2 min of the recording at baseline ICP. Gray highlighted regions in CPP show intact autoregulation. Figure 2a shows the relative arterial blood volume fraction, rV_a_. We found that rV_a_ monotonously decreased by 50% in the CPP range of 60–120 mmHg, indicating arterial vasoconstriction, a typical vasomotor response expected during CA. In this range, rCBF (Figure 2b) was maintained at a steady value with a nominal change of 10%. As CPP reduced below the lower limit of autoregulation (<~60 mmHg), rV_a_ started decreasing, which also caused a decrease in rCBF. In contrast, for CPP values above the upper limit of CA (>~120 mmHg), rV_a_ started increasing, also increasing rCBF by almost 250%. The trends seen in rCBF have been reported before as the ‘Lassen’s Curve’ and is the foundation of static autoregulation studies [1]. rSaO_2_ (Figure 2c) and rStO_2_ (Figure 2d) remained largely constant throughout, albeit showing a 5% change during CA andupto 10% at extreme CPPs.

The estimated rOEF_GM_ (Figure 2e) also remained largely constant within the region of CA but decreased at extreme CPPs by ~15%. rCMRO_2 GM_ (Figure 2f) closely mimicked the trends in rCBF, indicating a tight coupling between the two for most CPP values.

### 2.2. Comparison with Other Models

Among different models proposed for rCMRO_2_ and rOEF estimation, some of the most commonly used assumptions are a constant arterio-venous fraction (γ) and a constant SaO_2_ (rSaO_2_ = 1) [9,13]. We refer to this model here as the “constant γ” model (see Section 4.5). Using our experimental data, we contrast the rCMRO_2_ and rOEF estimated by the “constant γ” model with those obtained using the aforementioned “general model”, specifically as a function of CPP.

The percent difference in the rOEF (rOEF_GM_ vs. rOEF_const. γ_) and rCMRO_2_ (rCMRO_2 GM_ vs. rCMRO_2 const. γ_) calculations are shown in Figure 3a,b respectively. The discrepancy between the two for each, rOEF and rCMRO_2_, was calculated at every CPP value. Finally, the average and standard error for all points over 2 mmHg was reported. There was ~ 5% discrepancy between the two models for a CPP range of 60–120 mmHg, approximated as the region of CA in Figure 2. However, above and below this range, the discrepancy increased to 20%.

The primary factors contributing to these differences in the estimation by the two models are the changes in rSaO_2_ (Figure 2c) and the arterio-venous fraction γ (Figure 3c) with CPP. The latter parameter, γ, was evaluated using Equation (7) (see Section 4.5) and plotted as a function of CPP (Figure 3c). The trends in γ with CPP inversely followed the relationship between rV_a_ and CPP as expected from their mathematical relationship. Despite normalization by venous volume changes, γ shows a ~10% change with CPP.

### 2.3. Metric for Autoregulatory Assessment

We report changes in various cerebral tissue oxygenation and metabolic metrics (Figure 2) with CPP and CA, in an attempt to highlight the ability of multi-modal tissue spectroscopy to act as a non-invasive alternative for clinical assessment of CA. We plotted the rStO_2_, rOEF_GM_, and rCMRO_2 GM_ as a function of rCBF, color-coded by the CPP value at each bin (Figure 4a), to test the coupling between these different cerebral tissue oxygenation parameters. The most used clinical metric of CA failure is the uncoupling between CMRO_2_ and CBF. As expected, rCMRO_2 GM_ and rCBF showed a strong linear dependence, with a Pearson’s r of 0.84. rStO_2_ also increased linearly with CBF, with a Pearson’s r of 0.76. rOEF had the lowest Pearson’s r of −0.15 with CBF across all CPP values.

We further calculated the ratio between each of these metrics and rCBF and reported the median values and the distribution for three groups: within CA (CPP between 70 and 100 mmHg), below the lower limit of CA (CPP < 55 mmHg), and above the upper limit of CA (CPP > 135 mmHg) (Figure 4b) to find a clinically usable set of metrics that could indicate impaired CA. Here, the CPP ranges for each distribution were selected as a subset of those reported in the previous sections to avoid CA to dysregulation transitionary phases when estimating the metrics in the three groups. The ranges used here have also been previously reported as CA and dysregulation in neurocritical care [3].

A coupling ratio (rCMRO_2_/rCBF) of 1 in the region of CA indicates a healthy brain, where the supply and demand are well matched. From our experimental data, we observed a median coupling ratio of 0.98 in the region of CA (70–100 mmHg) and a significant decrease in this ratio at CPP below 55 mmHg (*p*-value << 1 × 10^−4^) and above 135 mmHg (*p*-value = 0.02), corresponding to the limits of CA. The ratio of rStO_2_ and rCBF was also ~1 in the region of CA, indicating a strong coupling. However, the ratio significantly (*p*-value << 1 × 10^−4^) increased at CPP < 55 mmHg and decreased for CPP > 135 mmHg, indicating a mismatch between the two parameters. rOEF/rCBF also significantly (*p*-value << 1 × 10^−4^) decreased with an increase in CPP. For all the metrics, significance was calculated using a Wilcoxon rank sum test (*ranksum*, MATLAB 2019b) between the CA group and the non-CA groups.

## 3. Discussion

Using simultaneous non-invasive tissue spectroscopy techniques (NIRS and DCS), we recorded hemodynamic signals while altering CPP using ICP changes in non-human primates (n = 4). Using these optical methods, in conjunction with invasive cerebral pressure sensing, we estimated rOEF and rCMRO_2_ using a general model to show their link to CA health. Here, we discuss the implications of our results and observations.

### 3.1. Tissue Oxygenation and Metabolism: Clinical Metrics for Diagnosing Autoregulatory Failure

A mismatch between cerebral oxygen supply and demand is one of the primary biomarkers of CA failure. A timely diagnosis and clinical intervention during such events can greatly improve patient prognosis. Usually, the ratio between rCBF (supply) and rCMRO_2_ (demand), also known as the coupling ratio, is used for such diagnosis. Our experimental data shows a strong coupling between rCMRO_2_ and rCBF as indicated by their strong linear dependence across CPP values (Figure 4a). However, the coupling ratio rCMRO_2_/rCBF is close to 1 (median value for CPP 70–100 mmHg is 0.98) only in the region of CA, indicating that rCMRO_2_ and rCBF are well matched. The value of the ratio significantly reduces beyond the limits of CA, showing a mismatch between the two parameters (Figure 4b).

However, the coupling ratio is a simplistic model for the supply–demand equilibrium in healthy cerebral tissue. Cerebral health also depends on the amount of oxygen extracted (rOEF), StO_2_, total blood volume, capillary density, hematocrit concentration, underlying neuronal activity, etc. [19]. A multi-modal setup, as in our experiments, allows for evaluation of these parameters and provides a more holistic cerebral health biomarker. Hence, we also tested how rOEF_GM_ and rStO_2_ coupling with rCBF changed with CA health. During hyperperfusion at high CPP, rOEF reduces in response to an excess of rCBF, which in turn decreases the coupling ratio [6]. In our work, we observed this decrease in rOEF (Figure 2e) above the limits of CA and in the reduction in rOEF/rCBF from its healthy value (median value for CPP 70–100 mmHg is 0.82) at high CPP (Figure 4). Conversely, during ischemic events at low CPP, rOEF is known to increase in response to the decrease in rCBF [13,20]. Contrarily, prolonged durations of CBF reductions during ischemic events can lead to a decrease in rOEF and, consequently, a decrease in demand (rCMRO_2_) with neuronal death [21]. The latter is also observed in our experimental data at low CPP (Figure 2e). This makes characterization of low CPPs solely using rOEF and the rOEF/rCBF ratio hard and heavily dependent on the duration of the ailment. This is one of the reasons why a combination of metrics using multi-modal imaging is beneficial.

Measurements of total hemoglobin, StO_2_, and differences in oxy- and deoxy-hemoglobin have been previously used as proxies for rCBF [22]. In a healthy brain, cerebral tissue perfusion is well matched with the oxygen concentration of the tissue, implying a linear relationship between the rCBF and rStO_2_. This relationship is also seen in the strong linear correlation between rCBF and rStO_2_ in our data (Figure 4a). Additionally, in the region of CA, rStO_2_/rCBF has a median value of 0.91. However, as CPP reaches extreme values, drastic changes in the vessel diameter (rV_a_ and rV_v_), rSaO_2_, and/or total hemoglobin can lead to a mismatch between rCBF and rStO_2_. This can be seen as a significant increase in rStO_2_/rCBF at low CPP and a decrease at high CPP, away from 1 (Figure 4b).

Thus, rOEF_GM_/rCBF and rStO_2_/rCBF in addition to the coupling ratio rCMRO_2 GM_/rCBF provide additional evidence for CA impairment, allowing for better clinical decision making. In addition to these, simultaneous monitoring of neural activity using electroencephalography can further add to the diagnosing power of CA impairment non-invasively at the bedside.

### 3.2. Other Models and Validity of Assumptions

Measurement of cerebral autoregulatory health using cerebral metabolism due to underlying neural activity at rest or evoked by an external stimulus has been modeled extensively, each focusing on different imaging modalities [23,24,25]. However, models solely using tissue spectroscopy have gained clinical popularity due to their ability to non-invasively and continuously monitor at the bedside [9,26,27]. The small footprint of NIRS and DCS probes also make them suitable for clinical use. Additionally, our work here focuses on monitoring tissue oxygenation and metabolic changes at rest, further removing the inconvenience of requiring patient participation in functional tasks.

Here, we compare and contrast two different tissue spectroscopy-based mathematical models for rOEF and rCMRO_2_ estimation. The “constant γ” model assumes an unchanging arterio-venous fraction (γ) and a constant SaO_2_ when calculating CMRO_2_. The arterial blood volume (rV_a_) is known to change with CPP as blood vessels constrict and dilate to compensate for CPP variations. Additionally, this ability is known to alter at extreme CPPs [1,28]. This trend is also observed in our estimation of rV_a_ using experimental data (Figure 2a). Conversely, venous volume and outflow are linked to ICP and consequently affected in CPP-based pathophysiology [26]. In addition, the total blood volume might also change locally or globally with autoregulatory impairment, which might also alter arterial and venous volumes [27]. In all these scenarios, the volume fraction γ might alter significantly with CPP, as seen in our results (Figure 3c). In addition, SaO_2_ also changes in many cerebral pathologies and is often used as a clinical biomarker for cerebral tissue health [29]. In our experiments, we observed a ~10% change in rSaO_2_ with CPP (Figure 2c). It must be noted here that in the case of hypoxia/hypercapnia-driven CA failures, changes in the partial pressure of blood gases can cause a more significant change in SaO_2_ [30]. In our experiments, the subjects were ventilated and EtCO_2_ constantly monitored (Figure 1b). Though not a direct measure of arterial partial pressure, EtCO_2_ did not vary significantly (25–40 mmHg) [31] in our experiments as shown in Figure 1b, indicating a minimal effect on SaO_2_.

Given the sources of error in the aforementioned model, we propose the use of a “general model” proposed by Kocsis et al. [16], which accounts for changes in γ and SaO_2_ when estimating rOEF and rCMRO_2_. To contrast the two models, we calculated the percent discrepancy between the estimates as a function of CPP. As expected, the two models agreed well in the range of CPP when CA was intact but showed a ~20% discrepancy at extreme CPPs. It must be noted here that our calculation of SaO_2_ assumes that changes in oxygenated hemoglobin at the heart rate (ΔHbO|_HR_) arise from arterial pulsations only, thereby allowing us to estimate SaO_2_ using NIRS. However, errors could arise from under/overestimation of the DPF and the time resolution in the spectrogram, limiting the identification of rapid transient changes in SaO_2_, and thereby affecting rOEF and rCMRO_2_ calculations.

### 3.3. Limitations and Future Directions

This work studied cerebral autoregulatory failure with ICP-dominated CPP changes, using diffuse optical spectroscopy methods for hemodynamic monitoring. While our unique experimental and imaging setup enabled us to evaluate rCMRO_2_, rOEF, and other tissue oxygenation metrics over a wide range of CPP, some confounding factors remain that were beyond the scope of this work.

Arterial hematocrit concentrations are often obtained from regular blood draws. To keep our estimations focused on non-invasive spectroscopy, we assumed a constant arterial hematocrit concentration (rH = 1) across all CPP values. We tried to eliminate inter-subject hematocrit variabilities by reporting values relative to the baseline of the respective subject. However, previous studies have also shown some correlation in the changes in hematocrit with blood pressure variations [32] and ischemia [33,34], which could still act as a source of error in our rCMRO_2 GM_ estimation. In a clinical setting, this can be managed using the indicator dilution method to obtain exact hematocrit concentrations.

Additionally, with DCS, there is also a lack of an absolute CBF measurement, which necessitates evaluations of relative values. While, in principle, our method allows for absolute OEF, and CMRO_2_ estimation, we were limited by the lack of a measure of hematocrit (H) and absolute CBF, both requiring supplementary invasive procedures to obtain absolute values. However, there have been recent advancements trying to make such recordings non-invasive [35,36]. This becomes particularly important when trying to set thresholds for clinical diagnosis in the absence of a healthy baseline value. However, as our method allows for long-term monitoring, trends over time in relative values can be used to inform clinicians about patient prognosis.

Another source of bias in our observations could arise from the small cohort of only male non-human primate subjects used for the study. While some studies have shown variations in CBF and CMRO_2_ with the age and sex of the subject [37], there are also studies contesting such differences [38]. To verify the presence of any such differences under our experimental paradigm, it would require a larger cohort or, preferably, a transition to a large-scale clinical study in patients with autoregulatory failure.

Finally, practical limitations prevented us from co-localizing our NIRS and DCS recordings, thereby probing bilateral tissue regions with each device symmetrically placed on the frontal cortex. Interhemispheric similarities in the hemodynamics and metabolism in the frontal cortex have been reported before [39] and, hence, should not affect our results and conclusions.

## 4. Materials and Methods

### 4.1. Subjects and Experimental Protocol

Continuous cerebral tissue spectroscopy, intracranial pressure, and arterial blood pressure data were collected from four healthy male Rhesus Macaques (Macaca Mulatta; average weight 10.25 ± 2.37 kg; average age 7.79 ± 1.31 years). For the duration of the experiment, each subject was anesthetized with 15–25 μg/kg/h of fentanyl and <1% isoflurane gas. This combination of anesthesia has been previously shown not to impair the autoregulatory capacity in our subjects [40]. The subjects were also maintained under 0.1 mg/kg/h of Vecuronium Bromide paralytic. Post sedation, surgical procedures were performed to place an A-line in the carotid artery to monitor blood pressure, and craniotomies to place an ICP sensor (Precision Pressure Catheter, Raumedic Helmbrechts, Germany) in the parenchyma of the brain and a catheter (Lumbar catheter, Medtronic, USA) in the lateral ventricle. The catheter was connected to a saline reservoir whose position altered the ICP, as seen in Figure 1a. Both blood pressure and ICP were recorded at 100 Hz. The scalp was surgically retracted before performing the craniotomies. All spectroscopy sensors were placed directly on the skull. The subjects were ventilated at 12 breaths/min for the entire duration of the experiment.

For each subject, the initial stable ICP reading after sensor insertion was considered as the baseline ICP. Subsequently, ICP was altered in steps of ~5 mmHg from baseline to 30 mmHg. For 3 out of 4 of the subjects, ICP was also returned to ~9 mmHg between each increase to reduce the additive effects of constantly increasing ICP. ICP manipulation was carried out by raising (increase in ICP) or lowering (decrease in ICP) the saline reservoir to alter the hydrostatic pressure and consequently the saline flow in the catheter to the ventricle. At each ICP value, data was recorded for ~1.5 h. Mean arterial pressure (MAP) was calculated as the sum of 1/3rd systolic peak value and 2/3rd diastolic value of each cardiac pulse of the arterial blood pressure recorded by the A-line. CPP was estimated as the instantaneous difference between MAP and ICP. For each subject, the experiments lasted ~15–24 h. An example time trace of the induced changes in ICP, subsequent changes in CPP, and changes in end-tidal CO_2_ (EtCO_2_) for NHP 2 is shown in Figure 1b.

### 4.2. Hemodynamic Signal Acquisition

Cerebral hemodynamic signals were measured using near-infrared spectroscopy (NIRS) and diffuse correlation spectroscopy (DCS). A frequency domain NIRS system (Oxiplex TS, ISS Inc, Champaign, IL, USA) was used in a multi-distance configuration to obtain the absolute tissue hemoglobin concentrations and changes in concentrations [41]. Intensity values were recorded at 690 and 830 nm with source-detector distances of 1.3, 1.7, 2.2, and 2.8 cm at 50 Hz sampling. The NIRS intensities were calibrated on a phantom with known absorption and scattering properties before starting data collection from the subject.

An in-house built DCS system was used in conjunction with NIRS to measure relative changes in cerebral blood flow. Data was collected at 50 Hz with a source-detector distance of ~2 cm. More details on the DCS device design have been presented in a previous paper [42]. Both the DCS and NIRS probes were placed in the frontal cortex on opposite hemispheres to avoid crosstalk.

### 4.3. Signal Processing

Timeseries from NIRS, DCS, and the pressure sensors were time-aligned using an analog signal sent by an external trigger at the beginning of each ICP manipulation. The time-aligned signals were then individually processed in MATLAB (version R2019b, MathWorks, Natick, Massachusetts, USA).

The multi-distance approach was used to estimate the absorption (µ_a_) and the reduced scattering (µ_s_′) coefficient of the tissue probed by NIRS. A linear fit between source-detector distances (d) and the log of AC intensity (ln(d2×IAC)) and phase were used to evaluate µ_a_ and µ_s_′, respectively. Any source-detector pair with a large variance from the linear fit were removed while keeping at least 3 distances for the final fit [10]. All linear fits had a coefficient of determination (r2) ≥ 0.9. The estimated tissue properties were then used to calculate the absolute oxy-, deoxy-, and total-hemoglobin concentration. StO_2_ was calculated as the ratio of oxy-hemoglobin to total-hemoglobin concentration.

The measured tissue optical properties (µ_a_ and µ_s_′) were also used to estimate a differential pathlength factor (DPF) under the assumption of a semi-infinite medium [43] for the third source-detector distance (2.2 cm). To account for the effects of geometry on DPF [44], the intensities were calibrated on a phantom with a similar curvature as the subject’s skull before beginning measurements. Additionally, in all our experiments, the probes were directly placed on the skull, significantly reducing the effects of superficial layers in our DPF estimates [45]. A moving average of the DPF with a window of 2 min was calculated and the values were then used to solve for changes in oxyhemoglobin (Δ*HbO*) and deoxyhemoglobin (Δ*HbR*) using the modified Beer–Lambert law [11]. An average of the first 1 min of the recorded DC intensities, for every subject, was used as the baseline intensity (I_o_) to calculate the changes in light intensity (ΔI). Any linear trends in the intensities were removed before applying the modified Beer–Lambert law. To estimate *SaO*_2_, changes in Δ*HbO* and Δ*HbR* at the heart rate were used:(1)$$SaO_{2}=\frac{\Delta HbO{|}_{HR}}{\Delta HbO{|}_{HR}+\Delta HbR{|}_{HR}}$$

Δ*HbO* at the heart rate (Δ*HbO*|*_HR_*) was calculated from a spectrogram of the signal with a 2-min window and no overlap between the windows. Δ*HbO*|*_HR_* was defined as the maximum spectral power of the Δ*HbO* signal in each window, within a frequency range of 1–4 Hz. The same heart rate frequencies were then used to calculate the spectral power of Δ*HbR* at heart rate (Δ*HbR*|*_HR_*).

Autocorrelation of the light intensity changes, recorded by the DCS, was used to estimate a scaled diffusion coefficient (αDb), which has been shown to be related to cerebral blood flow (CBF) [12]. An autocorrelation (g_2_) curve was calculated with an integration time-window of 20 ms and an overlap of 18 ms. The mean g_2_ curves and the standard deviation for a 1-h recording were used to define a signal-to-noise (SNR) ratio for the 1-h measurement. The resulting g_2_ curves were weighted by the SNR and fit to an analytical expression to estimate αDb at a signal temporal bandwidth of a 50 and 500 Hz sample rate.

### 4.4. Estimation of Tissue Oxygenation Metrics: General Model

We used the general model (subscript GM) proposed by Kocsis et al. [16] to evaluate rOEF_GM_ and rCMRO_2 GM_. An in-depth discussion of the assumptions and derivations for the equations can be found in the aforementioned paper. Here, we discuss some of the key equations:(2)$$OEF_{GM}=\left(1-\frac{StO_{2}}{SaO_{2}}\right)\left(1+\frac{\left(v_{a}\times rV_{a}\right)+\left(h\times \frac{v_{c}}{2}\right)}{\left(v_{v}\times rV_{v}\right)+\left(h\times \frac{v_{c}}{2}\right)}\right)$$
where all parameters with subscript a, c, and v indicate the respective values in the arteries, capillaries, and veins and h is the ratio of hemoglobin between the capillary and arterial compartment. All variables with a prefix ‘r’ represent the quantity relative to the baseline value. $rV_{a}$ is defined as the relative arterial blood volume, which can be determined by solving a fourth-order equation described by Kocsis et al. [16]:(3)$$\left({\tilde{R}}_{c}v_{a}^{2}\right)rV_{a}^{4}+\left(-2{\tilde{R}}_{c}v_{a}{\tilde{v}}_{c}\right)rV_{a}{}^{3}+\left({\tilde{R}}_{c}{\tilde{v}}_{c}^{2}-R_{a}v_{a}{}^{2}-R_{v}v_{v}^{2}\right)rV_{a}{}^{2}+\left(2R_{a}v_{a}{\tilde{v}}_{c}\right)rV_{a}-\left(R_{a}{\tilde{v}}_{c}^{2}\right)=0$$

All the parameters in lower case were constants, where $R_{a},\;R_{c},\;R_{v}\;$ represent the baseline resistance fractions and $v_{a},\;v_{c},\;v_{v}$ are the baseline volume fractions in the different vessel compartments. Their respective values are reported by Kocsis et al. [16]:$${\tilde{R}}_{c}=\frac{r\Delta ABP}{rCBF}-R_{c}$$
where rΔABP was calculated as the difference between the systole and diastole of each cardiac pressure waveform recorded by the A-line relative to the average difference value in the first 2 min of the recording (baseline ΔABP). If needed, this can easily be substituted with a non-invasive continuous blood pressure monitor [46]. rCBF is the CBF recorded by DCS relative to the average CBF in the first 2 min (baseline CBF):$${\tilde{v}}_{c}=\frac{rHbT}{rH}-v_{c}$$
where rHbT is the absolute total hemoglobin concentration relative to the average concentration in the first 2 min of recording (baseline HbT). rH is the relative average local hematocrit, which is assumed here to be unchanging with CPP; hence:rH=1
where $rV_{v}$ is the relative venous volume, which depends on the inflow ($rV_{a})$ and is given by:(4)$$rV_{v}=\frac{1}{v_{v}}\left(\frac{rHbT}{rH}-v_{c}-v_{a}rV_{a}\right)$$

A single $OEF_{GM}$ value was calculated using the mean values of the parameters over 2-min time bins. All values of parameters less than or greater than 3 standard deviations from the mean were removed and then a mean value was calculated for the time-bin. When solving for *rV_a_* and *rV_v_*, only *rV_a_* values (0, 2] and $rV_{v}$ > 0 were considered because any values outside these ranges would be physiologically unlikely [16]. If any of the time-bins yielded a value outside these ranges, that time bin was rejected. Finally, all 2-min time-bins with CPP or arterial blood pressure changes greater than 15 or ICP changes greater than 5 were also rejected from the analysis.

The $OEF_{GM}$ was then used to calculate $rCMRO_{2}{}_{GM}$:(5)$$rCMRO_{2}{}_{GM}=rSaO_{2}\;rOEF_{GM}\;rCBF$$

### 4.5. Alternative Model: Constant γ Model

We tested our general model against an alternative specific model (subscript const. γ) previously discussed by Culver et al. [13]. Using this model, OEF can be defined as:(6)$$OEF_{const.\;\gamma }=\left(1-\frac{StO_{2}}{SaO_{2}}\right)\frac{1}{\gamma }$$
where γ represents the arterio-venous volume fraction, which was considered a constant. However, to test for variations in γ with CPP (Figure 3c), we can use Equation (2) from the general model and equate it to the expression above, which gives:(7)$$\gamma =\frac{1}{\left(1+\frac{\left(v_{a}\times rV_{a}\right)+\left(h\times \frac{v_{c}}{2}\right)}{\left(v_{v}\times rV_{v}\right)+\left(h\times \frac{v_{c}}{2}\right)}\right)}$$

Finally, using the assumption of a constant γ and a constant *SaO*_2_ of 95%, rCMRO_2 const. γ_ can be expressed as:(8)$$rCMRO_{2\;const.\;\gamma }=rOEF_{const.\;\gamma }\;rCBF$$

## 5. Conclusions

Through controlled pre-clinical experiments conducted across four non-human primates with cerebral tissue oxygenation and metabolism studied over a wide range of CPP from 40–160 mmHg, we validated the use of non-invasive bedside spectroscopy techniques in evaluating CA health. Additionally, we tested two mathematical models and recommend the use of a general model for *rCMRO*_2_ and *rOEF* estimation, especially when studying patients with extreme CPP or altered CA. Finally, we suggest some metrics that can be evaluated with such a multi-modal imaging setup that can together act as a powerful tool for continuously monitoring patients and ensuring timely diagnosis of CA failure without added surgical cost or inconvenience.

## Figures and Tables

**Figure 1 metabolites-12-00667-f001:**
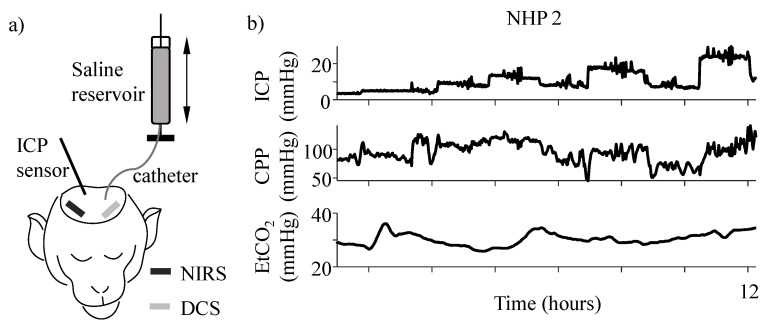
Experimental setup. (**a**) Schematic representation of the experimental setup. Black and gray bars at the frontal locations show the NIRS and DCS locations, respectively. ICP was measured using an invasive parenchymal pressure sensor and altered by changing the position of the saline reservoir connected to an intraventricular catheter. (**b**) Time traces of different experimental variables. The top panel shows the induced changes in ICP, the middle panel shows the calculated CPP (=MAP-ICP). The bottom panel shows a time trace of the recorded EtCO_2_ for the duration of the experiment.

**Figure 2 metabolites-12-00667-f002:**
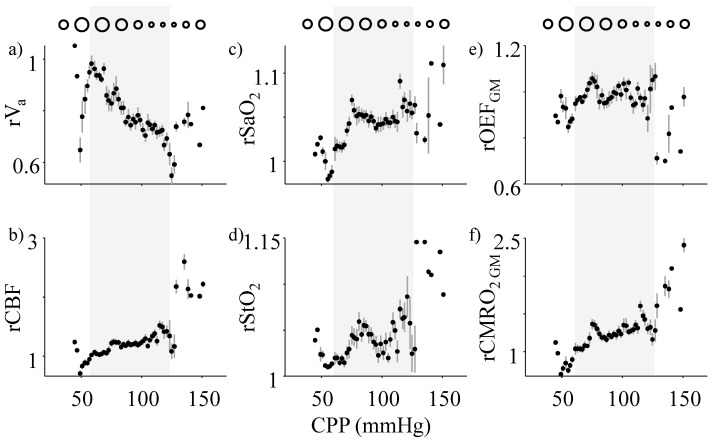
Trends in different oxygenation metrics with CPP. The circles at the top of each column show the expected trend of the vessel diameter changes with CPP. The different subfigures show trends in relative (**a**) arterial volume (rV_a_), (**b**) cerebral blood flow (rCBF), (**c**) arterial oxygen saturation (rSaO_2_), (**d**) tissue oxygen saturation (rStO_2_), (**e**) oxygen extraction fraction (rOEF_GM_), and (**f**) cerebral metabolic rate of oxygen (rCMRO_2GM_) as a function of CPP. The subscript GM is used for “general model” estimates. The approximate region of cerebral autoregulation is highlighted in gray across all panels. Every point on each graph indicates the mean value over a CPP bin of 2 mmHg and the gray vertical lines show ±1 standard error of the mean across all subjects.

**Figure 3 metabolites-12-00667-f003:**
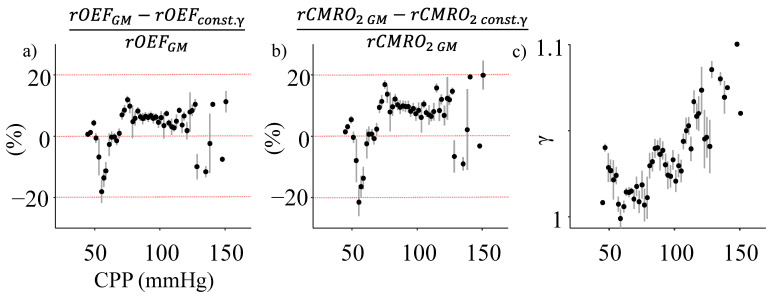
Estimation differences between the “general model” and the “constant γ” model. (**a**) The mean (black dot) and standard error (vertical gray lines) in the percentage difference in rOEF and (**b**) rCMRO_2_ is plotted against CPP. The red dashed horizontal line shows 0% and ±20% difference between the methods. (**c**) Changes in γ with CPP. All plots show data averaged across the four subjects.

**Figure 4 metabolites-12-00667-f004:**
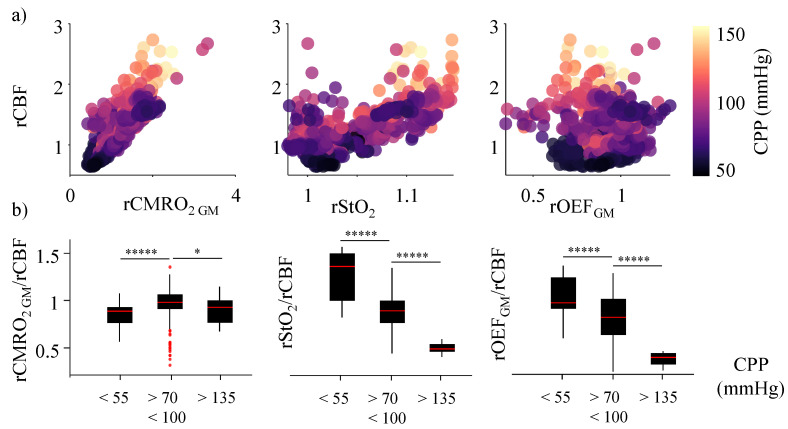
Use of tissue oxygenation metrics as an indicator of autoregulatory health. (**a**) Trends in estimated rCMRO_2 GM_, rStO2, and rOEF_GM_ as a function of recorded rCBF. A 2-min average of the data from all 4 subjects is shown as a point on the plot, color-coded by the average CPP (in the 2 min) at which it was recorded. The color scales from black (low CPP of ~40 mmHg) to light yellow (high CPP of ~160 mmHg). (**b**) The ratio of rCMRO_2 GM_, rStO2, and rOEF_GM_ to rCBF was calculated and split into 3 groups indicating impaired autoregulation: below the lower limit of autoregulation (<55 mmHg) and above the upper limit of autoregulation (>135 mmHg), and intact autoregulation (70 < CPP < 100 mmHg). Red horizontal lines indicate the median and the black solid box represents the 25th and 75th percentile of each group for the metrics. The red dots indicate the outliers for each group and the vertical line shows the spread of the respective distributions. The difference between two groups is shown by the horizontal line, with * to denote the significance. The *p*-value for all the indicated groups with five * was < 1 × 10^−4^ and for the group with a single * was 0.02.

## Data Availability

The data and associated codes presented in this study are openly available through FigShare under the https://doi.org/10.6084/m9.figshare.c.6049967, accessed on 14 July 2022.

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
