# Peer review of "Non-Invasive Spectroscopy for Measuring Cerebral Tissue Oxygenation and Metabolism as a Function of Cerebral Perfusion Pressure"

_metabolites, 2022, doi:10.3390/metabo12070667_

Round 1

Reviewer 1 Report

In this manuscript the authors describe a non-invasive spectroscopy technique to measure the oxygenation of brain tissue. The drafting of the manuscript is good, the introduction and discussion are exhaustive and inherent to the purpose of the work. The results and materials and methods clearly described. Understanding the difficulty of the type of experiment the only note is the small number (4) of primates, in my opinion there are few to deduce a significant results.

Explain the reason for the small number of primates used in the study

Author Response

We would like to thank the reviewer for their positive feedback and assessment of our work. We greatly appreciate their feedback and comments.

We understand and recognize the concerns raised by the reviewer regarding the small sample size of n=4. We would like to highlight that these were acute, terminal experiments which ran for well over 24 hours per subject. While we report data from 4 subjects, each subject had multiple simultaneous NIRS and DCS recordings at many different CPP values. The strength of our conclusions come from the analysis, having an average of 250 individual CMRO2 and OEF estimations per subject, over a wide CPP range of 40-160 mmHg. However, we understand the limitations of our study and highlight it in our discussion-

“Another source of bias in our observations could arise from the small cohort of only male non-human primate subjects used for the study. While some studies have shown variations in CBF and CMRO2 with age and sex of the subject,[40] there are also studies contesting such differences.[41] To verify the presence of any such differences under our experimental paradigm it would require a larger cohort or, preferably, a transition to large-scale clinical study in patients with autoregulatory failure.”

Reviewer 2 Report

This manuscript presented an interesting work of using non-invasive NIRS to measure cerebral tissue oxygenation and metabolism as a function of cerebral perfusion pressure. This manuscript also demonstrated a new general mathematical model for improved cerebral physiology resolved from the measurement in comparison to that by a known model.

Several issues listed below shall be addressed, before this work could be accepted into publication. 

1. Figure 2. It is nice to have the "trend of vessel diameter changes with CPP" sketched atop (a). It will be nice to see the same sketch atop (c) and (e). 

2. Figure 3. (a) and (b) can have the (+/-10% error and) +/-20% error marked with lines parallel to the red dashed line corresponding to 0%.

3. Figure 4, (a). Is there an expected trend between the two parameters shown in each sub-figure? Can you add a line to show that trend? 

4. Section 4.3 Signal processing. (a more problematic point)

"The multi-distance approach was used to estimate the absorption (μa) and the reduced scattering (μs´) coefficient of the tissue probed by NIRS. A linear fit between source-detector distances and the log of DC intensity as well as phase were used to evaluate μa and μs´, respectively.".

I believe that you will need to present exemplary figure to show the "linear fit". As your NIRS operates with source-detector distances that are likely in the semi-diffusive and diffuse region, there might not be a linear fit between source-detector distances and the log of DC intensity even in ideal condition.

Lets use I to represent a DC or AC intensity, and d to represent the source-detector distance. Over the sub-diffusive region, you might find a linear fit between log(I*d) versus d, and over the diffuse region, you will likely find a linear fit between log (I*d^2) versus d.  Your fitting principle does not agree with either of these two cases that are known to me.

As all your results depend on this fit, it would at least be imperative to show that the linear-fit does represent the data. 

The phase does follow a linear fit with d, that is well known.

5.  Your method of signal processing is also correlated with the use of differential pathlength factor. You have straightforwardly used the DPF assuming a semi-infinite geometry, however, any under- or over-estimations that assumption could have introduced will need to be addressed, in the context of the DPF being dependent upon the surface shape of the tissue.

A quick pubmed search lists about 6 references, among those, the three specified below could be helpful to your discussion. 

https://pubmed.ncbi.nlm.nih.gov/?term=differential+pathlength+factor+geometry&sort=date

Effect of adipose tissue thickness and tissue optical properties on the differential pathlength factor estimation for NIRS studies on human skeletal muscle.

Pirovano I, Porcelli S, Re R, Spinelli L, Contini D, Marzorati M, Torricelli A.Biomed Opt Express. 2020 Dec 22;12(1):571-587. doi: 10.1364/BOE.412447. eCollection 2021 Jan 1.

PMID: 33659090 Free PMC article.

On the geometry dependence of differential pathlength factor for near-infrared spectroscopy. I. Steady-state with homogeneous medium.

Piao D, Barbour RL, Graber HL, Lee DC.J Biomed Opt. 2015 Oct;20(10):105005. doi: 10.1117/1.JBO.20.10.105005.

PMID: 26465613 Free PMC article.

Estimating the Dependence of Differential Pathlength Factor on Blood Volume and Oxygen Saturation using Monte Carlo method.

Chatterjee S, Kyriacou PA.Annu Int Conf IEEE Eng Med Biol Soc. 2019 Jul;2019:75-78. doi: 10.1109/EMBC.2019.8856437.

PMID: 31945848 

   6. Your analytical notations can be improved. For examples, the rVv when standing alone is OK, but when it is mixed with other symbols it becomes quite confusing. I would suggest that you use V, and put r as a superscript and keep v as the subscript. That way you have a single V that is specified by the superscript or/and subscript and it makes it clean when combined with other symbols.  Similar treatment can apply to other symbols, such as OEF_GM. 

Author Response

This manuscript presented an interesting work of using non-invasive NIRS to measure cerebral tissue oxygenation and metabolism as a function of cerebral perfusion pressure. This manuscript also demonstrated a new general mathematical model for improved cerebral physiology resolved from the measurement in comparison to that by a known model.

Several issues listed below shall be addressed before this work could be accepted into publication.

We thank the reviewer for such detailed feedback on our manuscript. We greatly appreciate the comments and hope our replies and changes address the concerns raised.

  1. Figure 2. It is nice to have the "trend of vessel diameter changes with CPP" sketched atop (a). It will be nice to see the same sketch atop (c) and (e).

Thank you for the recommendation. We have now added the vessel diameter sketch on the top of all the panels. We have also attached the figure here –

  1. Figure 3. (a) and (b) can have the (+/-10% error and) +/-20% error marked with lines parallel to the red dashed line corresponding to 0%.

We agree with the reviewer that adding lines is helpful and have now added a line to indicate ±20% error. For the sake of neatness, we have left out the ±10% error line. The figure is also attached here-

  1. Figure 4, (a). Is there an expected trend between the two parameters shown in each sub-figure? Can you add a line to show that trend?

The reviewer raises an excellent point. While we do not explicitly draw-out the trend between the different variables, we do elaborate on each of them in our discussion and results. While the often-studied trends between CBF and CMRO2 and StO­2 follow a linear trend, trends with OEF are more nebulous. For consistency’s sake, we left out drawing an ideal trend. In addition, we show data point from across the entire CPP range of 40-160 mmHg across all three subjects, where changes in linearity (slope and intercept) can happen with CPP. To avoid binning the data for drawing “ideal” trends (which we do for the part b of the figure), we just show all the points and guide the reader through them in-text.

We hope this justifies our choice and that the reviewer understands our decision to not show “ideal” trends in the figure.

  1. Section 4.3 Signal processing. (a more problematic point)

"The multi-distance approach was used to estimate the absorption (μa) and the reduced scattering (μs´) coefficient of the tissue probed by NIRS. A linear fit between source-detector distances and the log of DC intensity as well as phase were used to evaluate μa and μs´, respectively."

I believe that you will need to present exemplary figure to show the "linear fit". As your NIRS operates with source-detector distances that are likely in the semi-diffusive and diffuse region, there might not be a linear fit between source-detector distances and the log of DC intensity even in ideal condition.

Lets use I to represent a DC or AC intensity, and d to represent the source-detector distance. Over the sub-diffusive region, you might find a linear fit between log(I*d) versus d, and over the diffuse region, you will likely find a linear fit between log (I*d^2) versus d.  Your fitting principle does not agree with either of these two cases that are known to me.

As all your results depend on this fit, it would at least be imperative to show that the linear fit does represent the data.

The phase does follow a linear fit with d, that is well known.

We thank the reviewer for their comment and apologize for the oversight on our part to correctly report the y-variable for the linear fits. For our work, linear fit between the ln (d2*IDC) and d (where d is the source-detector distance) were estimated. We have now corrected that in our reporting of the methods –

“The multi-distance approach was used to estimate the absorption (µa) and the reduced scattering (µs´) coefficient of the tissue probed by NIRS. A linear fit between source-detector distances (d) and the log of AC intensity (ln(d2IAC)) as well as phase were used to evaluate µa and µs´, respectively.”

The linearity between the (ln(d2IDC)) and d is an approximation often used for estimating µa and µs’ especially when using source-detector distances around ~1.5 cm or greater - [https://doi.org/10.1088/0031-9155/44/6/308] and [https://doi.org/10.1364/JOSAB.11.002128]. We did not specifically comment on the use of linearity for our fits as this has been widely used in previous literature. However, we have now added a line to indicate the goodness of fits for our data –

“Any source-detector pair with a large variance from the linear fit were removed, while keeping at least 3 distances for the final fit.[10] All linear fits had a coefficient of determination (r2)  0.9.”

  1. Your method of signal processing is also correlated with the use of differential pathlength factor. You have straightforwardly used the DPF assuming a semi-infinite geometry, however, any under- or over-estimations that assumption could have introduced will need to be addressed, in the context of the DPF being dependent upon the surface shape of the tissue. A quick pubmed search lists about 6 references, among those, the three specified below could be helpful to your discussion.

“Effect of adipose tissue thickness and tissue optical properties on the differential pathlength factor estimation for NIRS studies on human skeletal muscle.”

“On the geometry dependence of differential pathlength factor for near-infrared spectroscopy. I. Steady-state with homogeneous medium.”

“Estimating the Dependence of Differential Pathlength Factor on Blood Volume and Oxygen Saturation using Monte Carlo method.”

The reviewer points out some excellent considerations necessary when estimating DPF. We are aware of and accounted for the effects of changes in geometry by calibrating our intensities and probe before beginning measurements on a phantom with a similar curvature as the subject skull. The source-detector distances were calculated with this curvature accounted for. Additionally, in all our experiments, the probes were directly placed on the skull, significantly reducing the effects of superficial layers in our DPF estimates. We believe these measures greatly reduced the chances of error and potential over/under-estimation of DPF.

Finally, we would like to clarify here that the reason for estimating DPF in our case was to increase precision of the estimates and take into consideration changes in DPF (rather than absolute values) due to drastic changes in blood volume with CPP. We also performed the analysis with constant literature values for DPF for each wavelength and that did not alter the results and conclusion in any significant way. This finally points to the small contribution of DPF in our relative CMRO2 and OEF estimates and hence unlikely to alter results.

  1. Your analytical notations can be improved. For examples, the rVv when standing alone is OK, but when it is mixed with other symbols it becomes quite confusing. I would suggest that you use V, and put r as a superscript and keep v as the subscript. That way you have a single V that is specified by the superscript or/and subscript, and it makes it clean when combined with other symbols. Similar treatment can apply to other symbols, such as OEF_GM.

We thank the reviewer for their comment and understand that the current notation has a lot of qualifiers. We have followed notations to indicate relative changes by using r, as often seen in the DCS/NIRS literature when referring to relative CBF (rCBF). Hence, we believe the notation of rVv is less confusing than a subscript. But we do agree that this is not always easier to understand, and we did identify an equation where the prefix r for relative might become particularly confusing. We have now changed equation (3) from the paper which was –

To this –

  (3)

All the parameters in lower case were constants where  represent the baseline resistance fractions and  are the baseline volume fractions in the different vessel compartments.

We hope our explanation of the terms and qualifiers is sufficient to guide the reader through the paper.

Round 2

Reviewer 2 Report

This revision has included changes responding to most of concerns raised by this reviewer.

The comments 1-4 of this reviewer have been adequately and appropriately addressed by correcting or revising the pertinent materials.

In terms of the comment #5, your response in the cover letter could be embedded in the corresponding section (4.3) by also citing a few more references.    

Author Response

We have now added a commentary on errors in DPF calculation and its effect on our estimates both in the Discussion-
"To contrast the two models, we calculated the percent discrepancy between the estimates as a function of CPP. As expected, the two models agreed well in the range of CPP when CA was intact but showed a ~20% discrepancy at extreme CPPs. It must be noted here that our calculation of SaO2 assumes that changes in oxygenated hemoglobin (ΔHbO) at the heart rate arise from arterial pulsations only, thereby allowing us to estimate SaO2 using NIRS. However, errors could arise from under/over-estimation of the DPF and the time resolution in the spectrogram, limiting the identification of rapid transient changes in SaO2, thereby affecting rOEF and rCMRO2 calculations."

and in the Methods -
"The measured tissue optical properties (µa and µs´) were also used to estimate a differential pathlength factor (DPF) under the assumption of a semi-infinite medium [46] for the third source-detector distance (2.2 cm). To account for the effects of geometry on DPF [47], the intensities were calibrated on a phantom with similar curvature as the subject’s skull before beginning measurements. Additionally, in all our experiments, the probes were directly placed on the skull, significantly reducing the effects of superficial layers in our DPF estimates.[48]"

The added text is shown in blue here.